# How the AHR Became Important in Cancer: The Role of Chronically Active AHR in Cancer Aggression

**DOI:** 10.3390/ijms22010387

**Published:** 2020-12-31

**Authors:** Zhongyan Wang, Megan Snyder, Jessica E. Kenison, Kangkang Yang, Brian Lara, Emily Lydell, Kawtar Bennani, Olga Novikov, Anthony Federico, Stefano Monti, David H. Sherr

**Affiliations:** 1Department of Environmental Health, Boston University School of Public Health, Boston, MA 02118, USA; wangzhy@bu.edu (Z.W.); hengyang@bu.edu (K.Y.); erlydell@bu.edu (E.L.); 2Graduate Program in Genetics and Genomics, Division of Graduate Medical Sciences, Boston University School of Medicine, Boston, MA 02118, USA; mesnyder@bu.edu; 3Department of Pathology and Laboratory Medicine, Boston University School of Medicine, Boston, MA 02118, USA; jkenison@bu.edu; 4Department of Environmental Health, Boston University, Boston, MA 02118, USA; blara10@bu.edu (B.L.); kbennani@bu.edu (K.B.); 5Boston Medical Center, Boston, MA 02118, USA; Olga.Novikov@bmc.org; 6Division of Computational Biomedicine, Department of Medicine, Boston University School of Medicine, Boston, MA 02118, USA; anfed@bu.edu (A.F.); smonti@bu.edu (S.M.)

**Keywords:** aryl hydrocarbon receptor, AHR, kynurenine pathway, cancer

## Abstract

For decades, the aryl hydrocarbon receptor (AHR) was studied for its role in environmental chemical toxicity i.e., as a quirk of nature and a mediator of unintended consequences of human pollution. During that period, it was not certain that the AHR had a “normal” physiological function. However, the ongoing accumulation of data from an ever-expanding variety of studies on cancer, cancer immunity, autoimmunity, organ development, and other areas bears witness to a staggering array of AHR-controlled normal and pathological activities. The objective of this review is to discuss how the AHR has gone from a likely contributor to genotoxic environmental carcinogen-induced cancer to a master regulator of malignant cell progression and cancer aggression. Particular focus is placed on the association between AHR activity and poor cancer outcomes, feedback loops that control chronic AHR activity in cancer, and the role of chronically active AHR in driving cancer cell invasion, migration, cancer stem cell characteristics, and survival.

## 1. The First Hints of a Role for the AHR in Cancer: Carcinogenic Environmental AHR Ligands

Using environmental chemicals as probes, a hydroxylase “inducer” was first discovered by Poland and Glover in 1973 in what may have been one of the most important discoveries in toxicology [1,2,3]. By 1974, it was known that mice of disparate genetic backgrounds exhibited different sensitivities to the environmental chemical 2,3,7,8-tetrachlorodibenzo(p)dioxin (TCDD) and that these differences were likely due to polymorphisms in this unidentified “induction” receptor [4,5,6]. With mounting evidence that TCDD is a type 1 carcinogen in animals and humans [7,8] came the realization that the carcinogenicity of at least some environmental chemicals might not require mutagenesis but might be by this postulated receptor. With the biochemical purification in 1988 of the aryl hydrocarbon receptor (AHR) [9], the sequencing in 1991 of the AHR’s highly conserved N-terminal sequence [10], and the cloning of the *AHR* gene in 1992 [11,12], came a better understanding of the AHR’s environmental ligand reactivity and its contribution to the induction of hydroxylases that generate mutagenic intermediates. In that vein, a variety of environmental chemicals, including polycyclic aromatic hydrocarbons (PAHs) [13,14,15], aromatic amines [16], and non-ortho-substituted planar polychlorinated biphenyls (e.g., PCBs-118, PCB-156, PCB-126) [17,18,19], were shown to act largely through the AHR, in some cases through ligand-induced, AHR-mediated transcriptional up-regulation of cytochrome P450 Phase 1 hydroxylases (CYP1A1, CYP1A2, CYP1B1) and biotransformation of the parent pro-carcinogen into mutagenic epoxide intermediates [20,21,22,23,24,25,26]. As would be predicted from this understanding of AHR activity, mice lacking these hydroxylases were shown to have a lower incidence of malignant lymphomas and other tumors induced, for example, by PAH [27,28,29]. Notably, it could have been worse. Functional changes in the *AHR* during human evolution resulted in lower reactivity to PAH, relative to non-human primates, and thereby lower sensitivity to toxic PAHs present in smoke while retaining sensitivity to biologically important nontoxic endogenous AHR ligands [30]. 

These findings provided a ready explanation for the association between the AHR and cancer. That is, nominally “resting” AHR was seen to be activated by environmental chemicals to induce Phase 1 P450 hydroxylases that generate mutagenic intermediates from the pro-carcinogen parent compounds or from some endogenous substrates (e.g., estradiol or polyunsaturated fats [31,32,33]). These intermediates mutate DNA and initiate cancer. As uncomplicated and attractive as that theory was, it still did not account for the carcinogenicity of TCDD, a non-genotoxic AHR ligand, or for the AHR-driven induction of a growing list of genes unrelated to chemical metabolism but clearly related to cancer (see Section 6.1, Section 6.2, Section 6.3 and Section 6.4). In addition, the seminal demonstration that the AHR is highly conserved throughout evolution [34,35,36] argued for some important function(s) for the AHR in normal cell physiology. Colloquially speaking, if starfish, sea urchins, arthropods, nematodes, mollusks, and fish express AHR homologues (as reviewed in [37,38]), then the AHR must be doing something important in mammals. Since many critical cellular functions in mammals were first identified in the setting of cancer in which malignant cells compromise these cellular functions, often by exaggerating them (e.g., growth, survival, migration), it should come as no surprise that some “nominal” AHR functions were first identified in the cancer context.

## 2. AHR Transcriptional Signaling

As summarized in a comprehensive review on AHR signaling [39], the AHR is a basic Helix-Loop-Helix-Per/ARNT/Sim (bHLH-PAS) family member and its canonical transcriptional AHR signaling pathway began to be defined in the early 1990s (Figure 1). During that period, “latent” AHR was shown to be confined to the cytoplasm and to exist in a complex with pp60 Src [40], Immunophilin-like Ah Receptor-interacting Protein (AIP) (also known as Hepatitis B virus X-associated Protein 2 (XAP2) [41]), two molecules of HSP90 [42], and the HSP90 co-chaperone, p23 [43]. Both HSP90 and p23 protect the AHR from degradation [44]. Furthermore, HSP90 blocks nuclear translocation and represses DNA binding [45]. Ligand binding to the PAS A and PAS B domains exposes a protein kinase C target site, the phosphorylation of which effects conformational changes and subsequent translocation of the complex to the nucleus. In the nucleus, HSP90, p23, and pp60 Src are released and the ligand-bound AHR complexes are released with the aryl hydrocarbon nuclear translocator (ARNT), first described in 1992 [46]. Domains within the AHR responsible for ARNT binding were identified in 1994 [47]. The C’ terminus of the AHR/ARNT heterodimer then binds to promoters bearing Aryl Hydrocarbon Response Elements (AHREs), also known as Xenobiotic Responses Elements (XREs) [48], and recruits a variety of co-activators including NCoA-2 and p/CIP [49]. Most transcriptionally-mediated AHR activities have been attributed to AHR binding to a consensus AHRE, 5′-(C)GCGTG-3′ [50]. However, alternative AHR complexes and corresponding DNA binding sites have been identified and are likely to be important in physiological AHR functions [51]. In this vein, the AHR can bind to the NF-κB subunits RelA [52] and RelB [53] and to KLF6 [54], enabling the AHR complex to bind to non-canonical “alternative” DNA binding sites. The AHR also signals through its binding to other receptors or transcription factors including, but not limited to, the estrogen receptor [55,56], E2F1 [57], Rb [58], and c-Maf [59]. Furthermore, the AHR contributes to intracellular signaling through non-transcriptional pathways. For example, the AHR associates with tissue factor (TF), preventing its degradation and enhancing thrombosis [60,61]. The AHR may also signal though its associated Src kinase [40,62,63,64] or through E3 ubiquitin ligase activity [65,66]. It is likely that this diversity of AHR-associated proteins, along with differential cofactor recruitment [67], is responsible for the extraordinary variety of AHR responses to a galaxy of endogenous, microbial, dietary, and environmental ligands (reviewed in [68]). Indeed, “normal” physiological AHR activity is involved in oocyte development [69,70], oculomotor development [71], blood vessel development [72,73], cardiomyocyte development [74,75,76], hematopoietic stem cell development [77,78], development of the intestinal immune system and the gut epithelial barrier [79], UVB repair responses in skin [80], and the development and function of a variety of immune cells (reviewed in [56,68,81]). 

## 3. Circumstantial Evidence: High AHR Expression in Many Cancers

Suspicions that the AHR plays some key role(s) in cancer arose from pioneering studies demonstrating dramatically increased AHR expression in numerous cancer subtypes, including Hodgkin’s lymphoma and chronic lymphocytic leukemia [82], adult T-cell leukemia [83], and cancers of the breast [52,56,84,85], head and neck [86,87], brain [88,89,90], kidney [91], lung [92,93], pancreas [94], and GI tract [95,96,97,98]. Increased AHR expression is so consistent in some tumor types that it has been proposed as a prognostic marker [91,99]. The demonstration that both malignant cells and stromal or fibroblast-like cells within the tumor microenvironment (TME) express high AHR levels, as first shown in 2000 [85] and later confirmed [100], suggested that the influence of the AHR in cancer may not be limited to acute induction of mutagenic intermediates and that the contribution of the AHR to cancer progression in the TME, as with most things AHR, may be complex and ongoing.

## 4. Evidence Builds: An Association between Chronic “Constitutive” AHR Activity and Cancer Patient Outcomes

In addition to being hyper-expressed in some malignant cells, the AHR was shown, as early as 2000 [85], to be “constitutively active” in adult T cell leukemias [83] and cancers of the stomach [101,102], liver [103,104], prostate [105], head and neck [86,87,106], breast [100,107,108,109], brain [88,90], and skin [110,111]. The use of the term “constitutive” in this context reflected the field’s former lack of understanding about endogenous ligands in the tumor, and not the absence of chronic production of endogenous ligands (see Section 5.1). As illustrated in Figure 2, this AHR activity is readily identified by nuclear AHR localization in bladder [112], cervical [113], brain [114], pancreatic [115], head and neck squamous [116], lung adeno [117], lung squamous [117] and skin [118] carcinomas. Increased nuclear AHR localization was positively correlated with a higher tumor grade, more poorly differentiated cells, and/or poor prognosis in prostate, oral, and breast cancers [99,105,106,119,120], suggesting that the AHR may be contributing to increasing cancer aggression. 

As would be expected, high AHR expression and activity was usually, although not always [87], correlated with up-regulated *CYP1A1* and/or *CYP1B1* in cancers of the GI tract, bladder, head and neck, and breast [84,106,119,121,122,123]. Studies in the 1990s suggested that activated AHR, in at least some cancers, predominantly induced *CYP1B1* rather than *CYP1A1* transcription [84,93,124,125,126], an outcome that could reflect the contribution of distinct AHR-associated proteins such as the AHR interacting protein (AIP) [127] or differential recruitment of co-activators. Indeed, AHR-driven CYP1B1 was proposed as a universal tumor marker that could be immunologically targeted with CYP1B1-derived peptide vaccines [128,129,130,131]. While AHR levels frequently correlated with increased tumor aggression [119], AHR levels per se were understood to not necessarily represent levels of AHR activity. To illustrate this point here, we developed a biomarker set of genes regulated by the AHR in cancer by transcriptional profiling triple negative MDA-MB-231 breast cancer cells and triple negative SUM149 inflammatory breast cancer cells after CRISPR/Cas9-mediated AHR deletion (Figure 3a) and correlated the set of genes down-regulated in both knockouts with patient survival. Using a q value (FDR adjusted *p* value) of ≤0.05 as a cutoff, we demonstrated that 644 genes were significantly down-regulated in both cell types as compared with matched cells transduced with Cas9 without a guide RNA (“Cas9 controls”) (Appendix A) (Figure 3b). Using this set of AHR-regulated genes as a biomarker set, Gene Set Variation Analysis (GSVA) [132] was used to test the association between the AHR biomarker set and patient survival using multiple TCGA data sets for the eight cancers shown in Figure 2**,** all of which express nuclear AHR. Higher expression in human cancers of the genes down-regulated by AHR knockout (i.e., those driven by baseline AHR activity) significantly correlated with poorer survival in all eight cancer types (Figure 3c). These new results add support to the hypothesis that increased AHR activity plays an important role in tumor aggression.

## 5. Regulators of AHR Activity

### 5.1. An AHR Amplification Loop: A Possible Driver of “Constitutively Active” AHR 

The demonstration of chronic AHR activity in a variety of cancers begged the question of what was persistently driving the AHR in the TME. While a limited number of rare AHR polymorphisms were identified, none appeared to play a key role in human cancer [133,134,135]. Historically, therefore, it has been assumed that the malignant cell itself and/or other cells in the TME produce endogenous AHR ligands that drive chronic AHR activity. Over time, various AHR ligands, including several members of the tryptophan metabolite pathway, were shown to be produced by malignant cells. For example, kynurenine (Kyn), a tryptophan metabolite in the dominant kynurenine pathway, was shown to be produced by breast, head and neck, and brain (glioblastoma) cancers at levels sufficient to activate the AHR (e.g., ~90 µM) [88,90,108] (Figure 4, black font). Note that Kyn should be considered to be an AHR ligand as well as a surrogate for downstream metabolites/AHR ligands, such as kynurenic acid [136], xanthurenic acid [108], cinnabarinic acid [137], and trace kynurenine derivatives [138], any of which could serve as the ultimate effector ligands in a given cancer type.

This production of Kyn and other tryptophan-derived AHR ligands is rate-limited by indoleamine 2,3,-diozygenase (IDO) and tryptophan dioxygenase (TDO) [139], hydroxylases that generate N-formyl-Kyn (Figure 4). N-formyl-Kyn is further reduced to Kyn followed by catabolism to kynurenic, xanthurenic, and cinnabarinic acids. *IDO* and *TDO* were first shown to be regulated by the AHR in macrophages and dendritic cells [140,141,142] and later in malignant cells [108]. Therefore, chronic AHR activity may be sustained in malignancies by AHR-regulated induction of IDO or TDO and production of Kyn pathway ligands in a positive amplification loop [108,143] (Figure 4). Since stromal cells in the TME may also express nuclear (i.e., active) AHR [85], non-malignant cells may contribute to this amplification loop. That said, it still is not clear what factor(s) primes the pump. It is possible that environmental, microbial, or dietary [30,144,145,146,147] AHR ligands, many of which can be detected in human sera or urine [148,149,150], initiate AHR activation and start a self-perpetuating AHR circuit. This would suggest that even transient exposure to environmental AHR ligands may initiate a domino effect that results in enhanced AHR activity and ends in malignant transformation. Conversely, it is possible that short-term exposure to AHR inhibitors may reset the amplification loop at a lower steady state. Indeed, bistable or multistable biological circuits, defined as positive or negative feedback pathways that can reset at high low, or intermediate activity states after perturbation with inhibitors or activators, have been described and modeled in various biological contexts [151,152,153,154,155]. For example, the level of the Cdc2-cyclinB/Wee1 signaling pathway or the strength of MAPK activity can be stably reset depending on the magnitude of the response to a perturbation [151]. This could have important implications for the consequences of even brief exposures to environmental AHR ligands and, conversely, for the use of AHR inhibitors as cancer therapeutics.

### 5.2. Negative Regulators of the AHR Amplification Loop 

In counterbalance to the AHR-IDO/TDO amplification circuit is a feedback pathway that limits AHR activity (Figure 4, red font). As early as 1999, it was known that the AHR transcriptionally induces its own inhibitory protein, the AHR Repressor (AHRR) [156,157]. Initially thought to work solely by competing with the AHR for its dimerization partner ARNT, the AHRR was subsequently shown to suppress AHR activity without affecting DNA binding [158,159]. Whatever the mechanism through which the AHRR represses AHR activity, low AHRR expression in breast, lung, stomach, cervical, and ovarian cancers (likely mediated by DNA hyper-methylation and gene silencing [160]) has led some to suggest that the AHRR is a “tumor suppressor” [160], a moniker consistent with the description of the AHR as a tumor promoter, at least in some cancers. Notably, low level AHRR expression in breast cancer is associated with poorer survival [100] and ectopic AHRR expression is associated with decreased invasion [107].

It also has been suggested that a second level of negative feedback within the AHR circuit is likely mediated by AHR induction of prototypic target genes such as *CYP1A1* and *CYP1B1*. In 2004, it was shown that pharmacological inhibition of CYP1A1 increased baseline AHR activity in rat hepatoma cells through the inhibition of CYP1-mediated catabolism of endogenous AHR agonist(s) [161]. To illustrate this point in cancer cells here, we generated CYP1B1 knockout SUM149 inflammatory breast cancer cells (Figure 5a) and quantified AHR activity in the presence or absence of AHR agonists using an AHR-driven (pGudLuc) reporter construct. The AHR knockout cells described in Section 4 were used as a positive control. AHR knockout significantly reduced baseline AHR reporter (pGudLuc) activity in naïve and DMSO groups (Figure 5b, first two green bars). CYP1B1 knockout enhanced baseline AHR activity (Figure 5b, first two red bars). Furthermore, AHR activity induced by several AHR ligands, including environmental (B[a]P) and endogenous AHR ligands 6-formylindolo[3,2-b]carbazole (FICZ), Kyn, or xanthurenic acid, was enhanced by CYP1B1 knockout (Figure 5, remaining red bars). These data illustrate the general conclusion in the field that the classic AHR-inducible hydroxylases likely participate in a negative feedback loop in cancer cells.

## 6. Consequences of Chronic AHR Activity in Cancer

### 6.1. AHR-Mediated Cell Migration and Invasion

The ability of malignant cells to migrate from a primary site into adjacent tissue and/or vasculature is a major determinant of metastatic potential [163]. One of the earliest studies implicating the AHR in tumor cell migration was published in 2005 and demonstrated that the migratory potential of immortalized mouse mammary fibroblasts was significantly decreased when the AHR was ablated, an effect likely due to the removal of the AHR’s stimulatory effect on the ERK-FAK-Rac-1 pathway [164]. Shortly thereafter, a second study leveraged the effects of environmental AHR ligands by demonstrating the pro-migratory effects of B[a]P and TCDD on breast cancer cells [165]. In a complementary approach, DiNatale et al. demonstrated that AHR inhibitors slowed the migration of oral squamous carcinoma cells [86]. Similar results were obtained in triple negative breast cancers cells, in which migration and anchorage-independent growth was diminished after AHR knockdown [166]. Our laboratory demonstrated that AHR inhibitors or AHR knockout slows triple negative breast cancer and oral squamous carcinoma cell migration [106,108,162]. Furthermore, ectopic IDO expression, excess Kyn, xanthurenic acid, pyocyanin (a bacterial AHR ligand), B[a]P, TCDD, and FICZ accelerate the migration of breast and/or oral cancer cells in an AHR-dependent fashion [106,108,162]. Similar trends have been seen in other cancers. For example, the AHR transcriptionally regulates Memo-1, a gene implicated in colorectal cancer migration [97]. AHR activation with TCDD induces *MMP-9* expression and gastric cancer cell invasiveness, an effect likely mediated through a c-Jun-dependent pathway [96]. TCDD and B[a]P also up-regulate MMP-9 in prostate cancer cells [167], while AHR knockdown decreases invasion of prostate cancer cells in matrigel [168].

### 6.2. AHR-Mediated Epithelial-to-Mesenchymal Transition (EMT) and Metastasis

Epithelial-to-mesenchymal transition (EMT) is a critical process during which epithelial cells lose apicobasal polarity, connective junctions, and the ability to bind to the basal lamina, which collectively leads to migration and metastasis [169]. E-cadherin is a central contributor to an epithelial morphology, although E-cadherin deficiency in and of itself is insufficient to induce metastasis [170]. E-cadherin expression is repressed by Snail family members Snail, Slug, and Twist, and their upregulation, along with intermediate filament proteins such as Vimentin, are markers of metastatic potential [169,171,172].

Perhaps the earliest and most convincing study implicating the AHR in EMT was provided by Brooks and Eltom in 2011 [173]. These investigators demonstrated that retroviral transduction of an *AHR* plasmid into non-transformed human mammary epithelial cells was sufficient to induce motility, migration, invasion in Matrigel, anchorage-independent growth, and markers of EMT including increased Vimentin and morphologic changes consistent with EMT. Consistent with these results, AHR hyperactivation with FICZ increased *Snai1*, *Twist1*, *Twist2*, and *Vim* expression and migration in triple negative breast cancer. These genes have two to five consensus AHR binding site sequences (5′-GCGTG-3′) in their promoter regions, suggesting direct transcriptional regulation [162].

High AHR expression also correlates with lymph node metastases and/or poor prognosis in inflammatory breast and esophageal squamous cell carcinomas (ESCC) [119,174]. In the ESCC context, AHR modulation with the partial AHR agonist 3,3′-diindolylmethane not only down-regulated Vimentin and Slug, but also inhibited the RhoA/ROCK1 pathway, which in turn suppressed COX2/PGE_2_ signaling, prostaglandin E2 production, migration, metastasis and EMT [174,175,176,177]. This appears to be a generalizable metastasis pathway in that both RhoA/ROCK1 and PGE2 have been implicated in lung and endometrial carcinoma metastasis [178,179,180,181].

Data from hepatocellular carcinomas (HCC) show a similar AHR effect, albeit through different signaling pathways. For example, AHR induction with a prototypic environmental AHR ligand and carcinogen, benzo[a]pyrene (BaP), induced long interspersed nuclear element-1 (Line-1) expression through TGF-α signaling [182], a known inducer of EMT and a facilitator of metastasis [100,183]. Given that Line-1 and other retrotransposon elements mobilize throughout the mammalian genome and damage host DNA via mutational insertions, these results suggest a wide-ranging effect of AHR activation on cancer progression to a highly metastatic state.

While efforts to understand how the AHR affects tumor migration, invasion, EMT and metastasis have focused on classic cancer progression-associated genes discussed above, one significant contributor could be one of the most obvious AHR target genes, *CYP1B1*, especially in hormone-driven cancers. Kwon et al. demonstrated that ectopic CYP1B1 expression enhances Wnt/β-catenin signaling, a driver of EMT [184,185], and increased invasion in MCF10A breast epithelial cells and/or ER^+^ MCF7 breast cancer cells at least in part by increasing Snai1, Twist1, and Vimentin and decreasing E-cadherin expression [186]. The effector of this EMT gene profile appears to be the transcription factor, Sp1, induced through CYP1B1-mediated estradiol metabolism. Similarly, our data indicate that CY1B1 knockout in ER^-^ breast cancer cells reduces *Wnt5b* expression and invasion in matrigel (data not shown).

The translational implications of all of these studies on migration and invasion is exemplified by the ability of non-toxic AHR inhibitors to completely block metastasis of cervical (HeLa), TNBC (MDA-MB-231), and OSCC (HSC3) metastasis in a zebrafish model [107], or by AHR knockout to block melanoma metastasis to the lung [187].

### 6.3. AHR Role in Cancer Stem Cell (CSC) Development

Perhaps the earliest indication that the AHR could be involved in attainment of stem cell qualities came in 2000 in a toxicology paper by Murante and Gasiewicz, which demonstrated that in vivo treatment with TCDD increased the percentage of bone marrow cells expressing phenotypic markers of hematopoietic stem cells (HSCs) [188]. Subsequent studies from the Gasiewicz group and others extended these results by demonstrating that the AHR plays a central role in HSC growth and differentiation [77,78,188,189,190,191,192] and in lineage commitment of bipotential (erythroid/megakaryocyte) stem cells [78]. Similarly, the AHR has been associated with normal embryonic stem cell function [76,193]. In a classic example of how basic toxicology leads to translational outcomes, follow-up studies demonstrated that an AHR inhibitor, Stemregulin-1, expands HSCs in vitro and that these expanded HSC populations shorten the recovery time in high dose chemotherapy-treated, stem cell-rescued cancer patients [194,195].

These studies are of relevance here given the importance of aberrant organ stem cells in cancer. Cancer stem cells (CSCs) are a relatively small population of chemo- and radio-resistant malignant cells that have the ability to self-renew and to generate progenitor cells that form the bulk of a tumor. CSCs commonly over-express normal organ stem cell-associated genes and have an increased propensity to invade, migrate and metastasize. A rapidly enlarging body of evidence implicates the AHR in these processes. For example, in head and neck carcinoma, lung carcinoma, and choriocarcinoma cell lines, the AHR regulates expression of an ABC transporter, ABCG2, which contributes to chemoresistance by exporting drugs out of the cell against a concentration gradient [86,196,197,198]. AHR expression is elevated and nuclear in choriocarcinoma [197], TNBC [162], and oral squamous cell CSCs [106]. Similarly, in oral cancer and triple negative or ER^+^ breast cancer cells, the AHR was shown to regulate aldehyde dehydrogenase (ALDH) [106,162,199], which, like ABCG2, is associated with chemotherapy export [200]. ALDH also is associated with increased tumor cell invasion, higher tumor grade, and poor survival [201,202]. Further, AHR hyper-activation with the endogenous ligand FICZ [148,203,204] induced migration and invasion-associated (*Snai1*, *Twist1*, *Twist2*, *Tgfb1*, *Vim*) and stem cell-associated (*Notch1*, *Notch2*, *Bmi1*, *Nanog*, *Sox2*, *Dppa3*) genes in triple negative ALDH^high^ breast cancer CSCs [162]. Promoters from 11 of these genes contain 3–13 consensus AHR binding site sequences and the AHR interacts directly with the *Sox2* promoter [162]. Wnt5a/β-catenin signaling also correlates with CSC phenotype and disease progression in inflammatory breast cancers [119].

Functional studies are consistent with the findings on AHR-regulated stem cell genes. Thus, as would be predicted from the studies on ABCG2 and ALDH1, AHR inhibitors increased the sensitivity of ER^+^ and ER^-^ breast carcinoma cells to Adriamycin and Paclitaxel [162,199], oral cancer cells to Cisplatin [106,199] and choriocarcinoma cells to Methotrexate [197], directly connecting AHR-driven CSCs to cancer treatment outcomes. With regard to these studies, the ability of some of the chemotherapeutics (e.g., Cisplatin) to reduce AHR signaling in and of themselves, may have contributed to the increased effectiveness of the combination of AHR inhibitor and chemotherapeutics [205]. That said, Paclitaxel and Adriamycin have been shown to enhance AHR expression in MDA-MB-231 breast cancer cells [206] and AHR activity in cardiomyocytes [206], results that one would have expected would lead to a reduction in efficacy of a chemotherapeutic plus AHR inhibitor regimen. The fact that the combination proved more and not less effective demonstrates how elusive it is in some contexts to predict the outcomes of AHR manipulation.

CSC-dependent low adherence spheroid formation of choriocarcinoma or breast carcinoma cells in vitro was suppressed with AHR inhibitors or AHR knockdown and increased with AHR agonists (e.g., TCDD) [162,197,199]. Importantly, AHR^high^/ALDH^high^ breast cancer CSCs were significantly more efficient at initiating tumors than AHR^low^/ALDH^low^ cells, and AHR knockdown with siRNA significantly reduced tumor-initiating capacity [162], a sine qua non of CSCs [207].

Finally, it is important to note that, in some instances, AHR down-regulation, not up-regulation, characterizes CSC maintenance. For example, decreased AHR signaling contributes to CSC maintenance in human acute lymphocytic leukemia [208]. This type of paradoxical result, elegantly reviewed in Murray et al. [209] and [210], is not unusual in the AHR field and is likely attributable to the ability of a variety of endogenous and exogenous ligands [211] to differentially recruit AHR co-factors in different tissue contexts effecting different outcomes [67]. Indeed, we have specifically addressed this paradox in invasion, migration, and in vivo metastasis assays in head-to-head comparisons between AHR agonists and inhibitors [107]. It was shown that AHR agonists TCDD and/or 3,3′-diindolylmethane and AHR inhibitors CH223191 [212] and CB7993113 [109] inhibited triple negative breast cancer invasion in vitro and metastasis in vivo. The ability of agonists to reduce these measures of tumor aggression is consistent with previous studies, for example, from Safe et al. [213,214,215] and Kolluri et al. [210,216], showing that AHR hyper-activation with agonists can be anti-tumorigenic. While the molecular signal that results in these apparently contradictory results is still unknown, it has been postulated that agonists such as TCDD, 3,3′-diindolylmethane, or Omeprazole [217] induce differential cofactor recruitment by the AHR then those recruited by endogenous ligands in cancer [107,218]. In essence, some agonists may “divert” the AHR from a pro- to an anti-tumorigenic signaling pathway.

### 6.4. The AHR’s Role in Malignant Cell Apoptosis

In addition to relatively unfettered growth, an increased ability to invade local tissue, and a propensity to migrate from the primary site, aggressive malignant cells possess the ability to survive what, to a normal cell, would be a lethal signal to initiate apoptosis. Relatively recent studies in cancer cells suggest that the AHR may play a role in that resistance to death. The AHR was first implicated in apoptosis control in 1996, with studies demonstrating that activated T cells were more susceptible to Fas-mediated death signals when exposed to TCDD [219]. In 2005, these studies were extended by Park et al., who showed that the AHR potentiates Fas-mediated apoptosis in hepatocytes [220]. Remarkably, attrition of primordial ovarian follicles, an apoptosis-mediated event responsible for normal ovarian germ cell development, was shown to be driven by the AHR, potentially explaining the fertility issues seen in women who smoke [221]. Although in these cases AHR had a pro-apoptotic effect, they demonstrated that apoptotic and AHR signaling pathways could be linked.

In cancer, the AHR appears to do the opposite, i.e., to suppress apoptosis. As noted in Section 6.3, AHR inhibition increases tumor cell susceptibility to chemotherapeutics [106,162,197,199]. As a corollary, AHR activation with TCDD inhibited apoptosis in lymphoma cells in vitro and in vivo through the induction of COX2 and dysregulation of BCL2 [222]. AHR-dependent COX2 induction was also shown to play a role in blocking apoptosis induced by UVB irradiation, Adriamycin, or the dual tyrosine kinase inhibitor Lapatinib in breast cancer cells [223], or in non-transformed, AHR-transfected breast epithelial cells [224]. AHR-mediated resistance to UVB radiation-induced apoptosis may be most relevant in skin photo-carcinogenesis [14,225], where it was shown, in 2013, that AHR desensitized keratinocytes to UVB-induced apoptosis signaling in consort with increased expression of E2F1 and CHK1 [226]. More recently, it was shown that the AHR also suppresses pyrimidine dimer repair in vitro and in vitro and blocks the formation of double strand breaks that lead to apoptosis [227]. Remarkably, AHR knockout mice exhibited 50% fewer UVB-induced cutaneous squamous cell carcinomas than wildtype mice [227]. Finally, Kyn increased expression of anti-apoptotic proteins cIAP-1, cIAP-2, XIAP and Bcl-2 and decreased pro-apoptotic Bax in a pancreatic cancer cell line [228].

## 7. Caveats: Interspecies Differences

While the studies summarized above begin to reveal commonalities in how the AHR influences carcinogenesis, some important caveats should be kept in mind, not the least of which is the interspecies differences between murine and human models. The murine AHR has approximately a 10-fold higher affinity for TCDD than the human AHR [229], a difference that suggests that humans may have a higher tolerance for TCDD than mice [229,230]. Furthermore, a relative lack of similarity of the carboxy (DNA-binding) terminus between the mouse and human AHRs may result in different co-factor recruitment, specifically LXXLL binding motifs, leading to distinct transcriptional activity [231,232]. Furthermore, using a transgenic model in which mice expressed the human AHR in liver, it was shown that the murine and human AHRs induce different transcriptional responses to a given ligand, including TCDD and endogenous indole compounds [229]. With regard to the latter class of agonists, the human AHR has a higher affinity for indirubin and generates a different transcriptional profile than the murine AHR, suggesting an evolutionary preference for endogenous ligands [30,233]. This species-specific ligand selectivity may explain many of the differences observed in human and murine models.

## 8. Conclusions

Here we have summarized a variety of studies, all of which rest on a foundation of toxicological research that was designed to define the basic molecular mechanisms through which environmental AHR ligands generate adverse biological responses. Many of the studies, conducted with diverse cancer types, exploited environmental and, more recently, endogenous AHR ligands to untangle the AHR’s role in cancer. The results clearly indicate a complex association between the AHR and several critical cancer features, including increased malignant cell invasion, migration, metastasis, CSC formation, and survival. For the most part, these studies have been conducted on a basic science level. However, the implications of these studies for our understanding of the genesis of many cancers, their prevention, and their treatment are far-reaching. Thus, these studies have lent credence to the argument that: (1) primary cancer prevention can be effected by minimizing exposure to subsets of environmental AHR ligands, (2) cancer interception can be considered prior to full blown malignancy if early markers of AHR activity can be identified (e.g., Figure 3), and (3) several cancers in which AHR levels correlate with poor survival may be treatable with specific AHR inhibitors.

## 9. Materials and Methods

### 9.1. Generation of AHR or CYP1B1 Knockout MDA-MB-231 and SUM149 Cell Lines with CRISPR-Cas9 Gene Editing

Human AHR and CYP1B1 knock-out SUM149 and MDA-MB-231 cell lines were created using lentiCRISPR v2 (Addgene no. 52961, Cambridge, MA, USA), which contains Cas9 and a guide RNA cloning site (*BsmBI*). Two target sequences (5′-CCTACGCCAGTCGCAAGCGG-3′ and 5′-CCGAGCGCGTCCTCATCGCG-3′) for AHR or the two target sequences (5′-TTAGCGGCCAAGGGTCGTTC-3′ and 5′-CCTGCTACTCCTGTCGGTGC-3′) for CYP1B1 knock-out were used. The target sequences are located in the first exon of the *AHR* and *CYP1B1* genes, respectively. Lentivirus particles were generated in the HEK293NT cells by co-transfecting the lentiCRISPR v2, or AHR-lentiCRISPR v2 plasmids, and the packaging plasmids (pLenti-P2A and pLenti-P2B, Cat. # LV003, Applied Biological Materials Inc. Richmond, BC, Canada), using Lipofectamine 2000 (Invitrogen, Grand Island, NY, USA), according to the manufacturer’s instructions. Virus-containing media were collected 72 h later and filtered through a 0.45 μm filter. SUM149 and MDA-MB-231 cells were transduced with lentiviruses in the presence of 5 µg/mL polybrene as described [234]. Forty-eight hours after transduction, cells were selected with 2 µg/mL Puromycin for 10 days. The efficiency of AHR and CYP1B1 knockout was validated by DNA sequencing and immunoblot analyses.

### 9.2. Western Blotting

Cells were lysed and protein extracted with RIPA (Radio Immune Precipitation Assay) buffer (Boston BioProducts, Ashland, MA, USA). Protein concentrations were quantified with a Bradford protein assay. Equal amounts of protein (30 µg) were subjected to 10% SDS-PAGE and then transferred to a nitrocellulose membrane. Non-specific binding sites were blocked with blocking buffer containing Tris-buffered saline and 0.1% Tween-20 with 5% nonfat milk powder for 1 h at room temperature, and the blot was incubated with 1:1000 dilution AHR- or beta-actin-specific antibody in blocking buffer. AHR antibody was purchased from Cell Signaling. β–Actin antibodies were from Sigma-Aldrich (St. Louis, MO, USA).

### 9.3. AHR-Driven Reporter Assay

SUM149 cells were co-transfected with the *pGudluc* reporter plasmid (0.5 µg) (generously provided by Dr. M. Denison, UC, Davis), and *CMV-green* (0.1 µg) (for normalization) using TransIT-2020 transfection reagent (Mirus, Madison, WI, USA). Transfection medium was replaced after 24 h. The cells were left untreated or dosed with vehicle (DMSO, 0.1% final concentration), 10 uM B[a]P, 0.5 uM FICZ, 100 uM kynurenine, or 100 uM xanthurenic acid and harvested after 24 h in Glo Lysis Buffer (Promega, San Luis Obispo, CA, USA). Luciferase activity was determined with the Bright-Glo Luciferase System according to the manufacturer’s instructions (Promega, Madison, WI, USA). Luminescence and fluorescence were determined using a Synergy2 multifunction plate reader (Bio-Tek, Winooski, VT, USA).

## Figures and Tables

**Figure 1 ijms-22-00387-f001:**
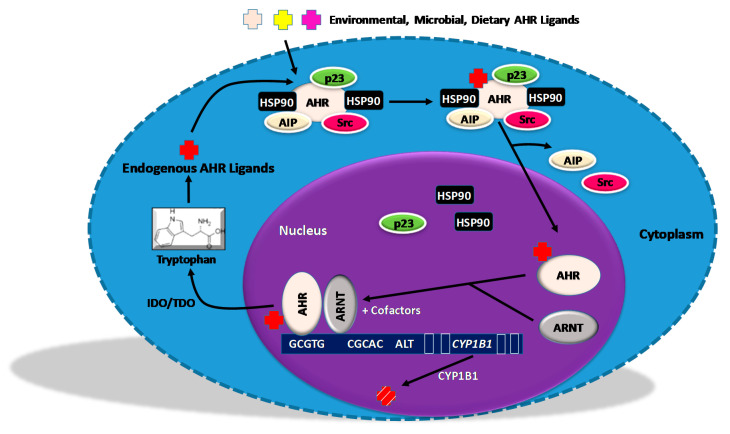
The Transcriptional AHR Signaling Pathway. Cytoplasmic aryl hydrocarbon receptor (AHR) exists in a complex with HSP90, Immunophilin-like Ah Receptor-interacting Protein (AIP), p23, and Src 40. Once engaged with exogenous or endogenous ligands, the AHR sheds AIP and Src and translocates to the nucleus. In the nucleus the AHR dimerizes with the Aryl Hydrocarbon Nuclear Translocator (ARNT), binds to consensus Aryl Hydrocarbon Response Elements (AHREs), recruits coactivators, and transactivates a battery of genes including the hydroxylases *CYP1B1* and *CYP1A1*, which metabolize some environmental AHR ligands into mutagenic epoxide intermediates. (Alternative AHR complexes containing NF-κB subunits, KLF6, or potentially other proteins bind to hybrid (AHRE/NFκB sites) or alternative DNA sequences to activate different sets of AHR responsive genes). CYP1B1 degrades at least some endogenous and exogenous AHR ligands in a negative feedback loop. The AHR complex can also induce IDO1/2 and/or TDO dioxygenases, which metabolize tryptophan into endogenous AHR ligands including, but not limited to, Kyn (kynurenine), in a positive feedback loop (see Section 5). Distinct sets of genes are activated by different AHR ligands, likely a result of differential co-factor recruitment. The AHR also functions through non-transcriptional pathways not represented here.

**Figure 2 ijms-22-00387-f002:**
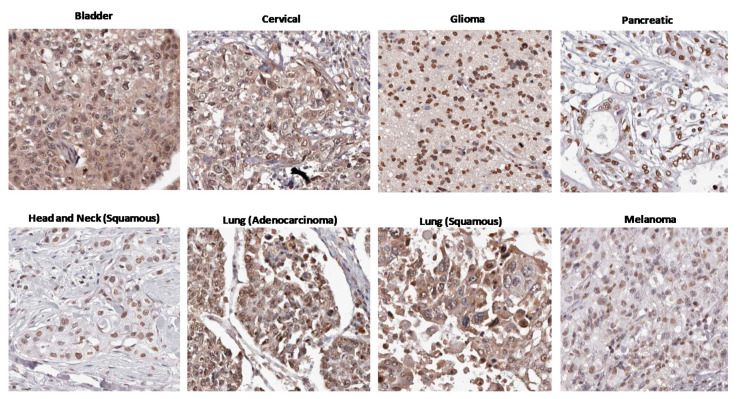
Nuclear AHR expression in eight human cancers. Immunohistochemistry images were obtained from the Human Protein Atlas. Nuclear staining for the AHR is evident in all eight cancer types.

**Figure 3 ijms-22-00387-f003:**
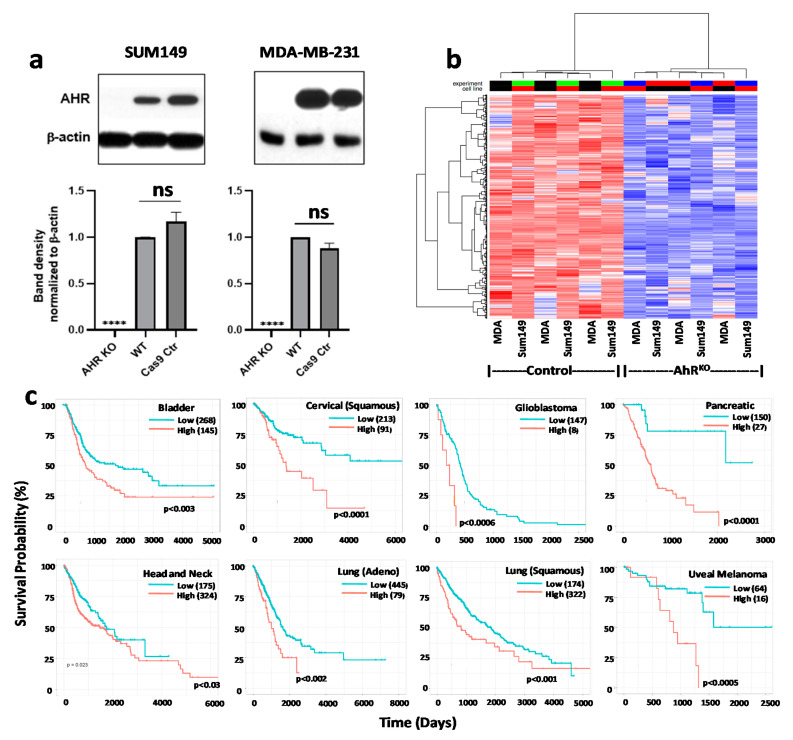
A 644 AHR-driven gene signature correlates with survival in eight human cancers. (**a**) Top: Western blotting for AHR protein in AHR knockout (KO), wildtype (WT) or Cas9 control SUM149 triple negative inflammatory human breast cancer cells or MDA-MB-231 triple negative human breast cancer cells. Bottom: Quantification of β-actin normalized band densities. Data are presented as means + SE from three independent experiments. No significant differences (ns) were found between WT and Cas9 (transfected with Cas9 but not guide RNA) control cells. **** *p* <0.00001 compared to WT or Cas9 controls. (**b**) Affymetrix whole human genome microarrays were used to determine transcriptional profiles of Cas9 control or AHR knockout SUM149 or MDA-MB-231 cells. Genes differentially expressed upon AHR knockout were identified across cell types. A significant decrease after AHR knockout was defined as an adjusted false discovery rate (*q* value) of ≤0.05. (**c**) The 644 AHR-driven gene signature in “b” was correlated with survival for the eight cancers in Figure 1 using TCGA data and Gene Set Variation Analysis (129).

**Figure 4 ijms-22-00387-f004:**
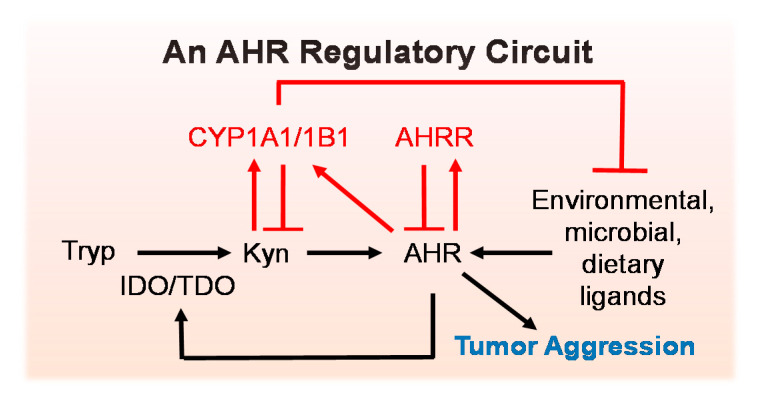
AHR Regulatory Circuit. Tryptophan is metabolized to Kynurenine (Kyn) and downstream AHR ligands that drive AHR activity and cancer aggression (migration, invasion, metastasis, “stem-ness”). Activated, nuclear AHR upregulates *Ido1*, *Ido2*, and/or *Tdo2*, producing more AHR ligands in an amplification loop that sustains AHR activity. AHR induces CYP1A1 and CYP1B1 (hydroxylases that can metabolize some AHR ligands) and AHR repressor (which limits AHR activity). Environmental, microbial or dietary AHR ligands may “prime the pump” and/or exacerbate the endogenous AHR signaling circuit.

**Figure 5 ijms-22-00387-f005:**
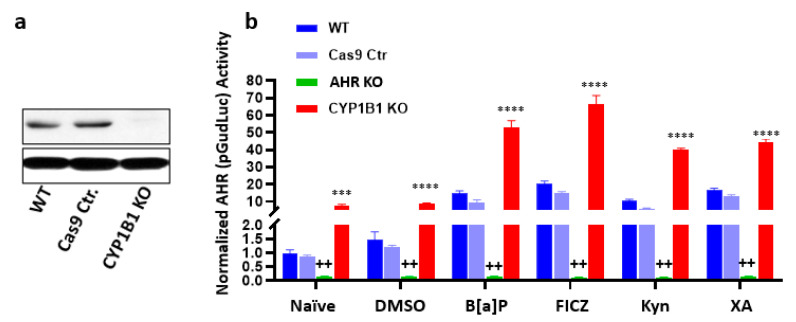
AHR Knockout Reduces and CYP1B1 Knockout Enhances Baseline and Ligand-induced AHR Activity. The AHR or CYP1B1 was deleted from triple negative, inflammatory human breast cancer cells [162]. (**a**) Western blotting showing the absence of detectable CYP1B1. (See Figure 3 for AHR knockout western blots). (**b**) CMV-driven GFP-normalized, AHR-driven pGudLuc reporter activity was assayed in wildtype, Cas9 control, AHR knockout, or CYP1B1 knockout SUM149 cells treated for 24 hours with 0.1% DMSO (vehicle), 10 uM B[a]P, 0.5 uM FICZ, 100 uM kynurenine (Kyn), or 100 uM xanthurenic acid (XA). Data are presented as normalized means + SE. from a minimum of three experiments. ^++^
*p* < 0.01 relative to similarly treated Cas9 control. *** *p* < 0.001 relative to similarly treated Cas9 control. **** *p* < 0.0001 relative to similarly treated Cas9 control.

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
