# Peer review of "How the AHR Became Important in Cancer: The Role of Chronically Active AHR in Cancer Aggression"

_ijms, 2020, doi:10.3390/ijms22010387_

Round 1
Reviewer 1 Report
The authors present a review that is focused on the evolution of the views of the role of Aryl Hydrocarbon Receptor (AhR) in cancer biology, from mediating toxicity of anthropogenic xenobiotics to disturbances of endogenous AhR activity leading to the cancer. Efforts to systematize knowledge about feedback loops that control AhR activity seem to be interesting. In general, the structure of the article needs some revision.
- The authors present their work as a review, but it contains the authors' experimental data: apparently Figure 2, 4 and 5, its description and discussion, chapter "Materials and Methods". This is rather within the competence of the Editor, but I am not aware of the practice of such a combination in one article. I think, the experimental part must be eliminated from the review and (if editorial policy allows) only be mentioned as unpublished data. In this regard, needs to be reorganize sections that are associated with this data.
- I would recommend writing an introduction and briefly describing the general aspects of AhR biology such as the signal transduction pathway, AhR ligands, non-cancer physiological role. The review objective needs to be clearly enunciated.
- Figure 1 requires a more accurate reference so that the original photos and their description, if any, can be easily found.
- Lines 163-166. Are there examples of similar events with any other molecular pathways where the known substance initiate a domino effect that results in enhanced pathology or reset the amplification loop? Please add this to the text if any.
- The first sentence of the abstract (Lines 25-27) looks inappropriate in a scientific paper. I would recommend to remove this.
Author Response
We would like to thank the reviewers for their careful analysis of our manuscript entitled “How the AHR Became Important in Cancer: The Role of Chronically Active AHR in Cancer Aggression” by Wang et al. We feel that the comments were fair and insightful and significantly helped us improve the manuscript. We have addressed the reviewers’ comments below. Again, thank you for your consideration and input.
Reviewer #1
The authors present a review that is focused on the evolution of the views of the role of Aryl Hydrocarbon Receptor (AhR) in cancer biology, from mediating toxicity of anthropogenic xenobiotics to disturbances of endogenous AhR activity leading to the cancer. Efforts to systematize knowledge about feedback loops that control AhR activity seem to be interesting. In general, the structure of the article needs some revision.
The structure has changed significantly. There are now two new sections (Sections 2 and 7), some of the primary data have been removed, and several paragraphs responding to the reviewers’ suggestions have been added.
- The authors present their work as a review, but it contains the authors' experimental data: apparently Figure 2, 4 and 5, its description and discussion, chapter "Materials and Methods". This is rather within the competence of the Editor, but I am not aware of the practice of such a combination in one article. I think, the experimental part must be eliminated from the review and (if editorial policy allows) only be mentioned as unpublished data. In this regard, needs to be reorganize sections that are associated with this data.
We have seen and written reviews that include primary data so there are precedents for this approach to a review paper. The editors of IJMS have agreed that inclusion of primary data is appropriate when it “…helps the flow of the review or reinforces the summary conclusions in the review”. Our intention was to use these data to help illustrate the main conclusions of the review. For example, Figure 2 assembles publically available data to illustrate the conclusion of many studies that the AHR is active (nuclear) in several cancers. Similarly, several published studies suggest that AHR activity is likely to have an adverse functional consequence in some but perhaps not all cancers. Figure 3, the presentation of survival data for patients with the cancers called out in Figure 2 and expressing higher levels of the AHR transcriptional profile, illustrates this position. That said, we take the reviewers point that it might be confusing to have both a review and primary data. Therefore, we have removed the original Figure 5 and have clarified when we are presenting primary data (e.g., lines 145-148). If the reviewer feels strongly about not having the primary data in this review, then we will gladly remove them.
- I would recommend writing an introduction and briefly describing the general aspects of AhR biology such as the signal transduction pathway, AhR ligands, non-cancer physiological role. The review objective needs to be clearly enunciated.
This is an excellent point and we now have added a new introduction (Section 2) and figure (Figure 1) describing AHR signal transduction.
We agree that the variety of responses from AhR ligands and the non-cancer physiological roles of the AHR are important, although each can be the subject of comprehensive review papers and are covered by other reviews in this special IJMS issue on the AHR. That said, we now have added a discussion of some non-cancer roles of the AHR in Section 2 (for example, lines 97-111).
We now have stated the objective of the review in the abstract. We also have clarified that the data in Figures 2, 3, and 5 are new data that illustrate the theme of the review, i.e., that that the AHR is up-regulated and active in some cancers.
- Figure 1 requires a more accurate reference so that the original photos and their description, if any, can be easily found.
We agree. We have now added the URLs for the Protein Cancer Atlas and for each of the histologies presented in the former Figure 1 (now Figure 2), including the specific slide in the Atlas that was selected, as references (references 112-118).
- Lines 163-166. Are there examples of similar events with any other molecular pathways where the known substance initiate a domino effect that results in enhanced pathology or reset the amplification loop? Please add this to the text if any.
Yes, there are instances where these kinds of biological circuits have been described and modeled. We now have added a brief description of those reports on lines 203-208.
- The first sentence of the abstract (Lines 25-27) looks inappropriate in a scientific paper. I would recommend to remove this.
The original lines 25-27 have been deleted as has a reference to those lines found at the end of the paper originally on lines 378 and 379.
Reviewer 2 Report
The paper by Wang et al „How the AHR Became Important in Cancer: The Roleof Chronically Active AHR in Cancer Aggression“ is, in a way, a non-traditional paper containing elements of both review and original research articles. Contrary to a usual review, it contains e.g. a section „Materials and Methods“. This hybrid nature is not a bad thing in itself, but it gives a reviewer challenge to keep these two parts separate.
A review of AHR is a most timely and needed event – over the last couple of years, many interesting original research papers have been published, which give a good reason for a review. In the context of AHR and cancer, there are couple of such reviews published recently (Baker at al Med Res Rev. 2020; Trikha and Lee, Biochim Biophys Acta Rev Cancer. 2020; Kolluri et al Arch Toxicol. 2017) and it would have been nice to see them mentioned by Wang et al.
AHR is a transcription factor and thus a very important part of cell signaling pathway. It is activated by ligands (exo- and endogenous), after which AHR translocates into nucleus and dimerizes with ARNT. It would be useful to start the review from this known molecular signaling pathway, because it is very central in explaining the functioning of AHR in both normal and cancer cells. Line 51 speaks about „induction of hydroxylases“ and line 55 about „AHR-mediated transcriptional up-regulation” - these expressions are simply not clear enough. And, after all, ARNT comes in as an AHR partner only at line 180, when the hypothetical model of AHRR action is described. So, my suggestion would be to start with Fig.1., which could explain the cellular AHR pathway.
A problem in this (and not only this) paper is that there is no clear distinction between results achieved with mouse models (and cells) and human ones. Our own experience shows that there are important and substantial differences between the two organisms and putting the data all together sometimes leads to a confusion.
Over the last two years, more than 100 papers have been published in the AHR field. Although the Wang et al review covers excellently the historical part of AHR research, which started in 1970-ies, it would be important to include also much more data from recent years. Along the same lines – a 2016 paper (Kwon et al) can hardly be called a recent one in a review, which will be published in 2021 (see line 279).
I have also some remarks concerning the original research results included in the paper. First, the Affimetrix data of two cell lines with knock-out AHR claim that there were 644 differences (down-regulations?) in gene expression between AHR+ and AHR- lines. For any standard of scientific publication, one would expect to see the list of these 644 genes at least as a Supplementary information or equivalent. Additional small technical problem is the use of cells transfected with Cas9 and no guide RNA as a control equivalent to wt. At Fig. 4a one can see that AHR expression is higher in Cas9 transfected cells that in wt, which itself can cause changes in expression levels of some of these 644 genes.
As correctly stated (lines 79 and further; 105 and others), higher AHR expression levels have been seen in many cancers. However, this is not always the case and there are many examples of the contrary. I find that this controversy would need some discussion, especially in the context of Fig.2.
The exception of Twist1 (line 258) in the list of the genes increasing their expression after AHR hyperactivation is not entirely correct – one can find 5’-GCGTG-3’ in both human and mouse Twist1 DNA sequences.
Line 324 and further: AHR inhibitors increase the sensitivity of breast carcinoma cells to Adriamycin and Paclitaxel and oral cancer cells to Cisplatin. It should be mentioned, however, that Adriamycin, Paclitaxel and Cisplatin also interact with AHR directly. Cisplatin is an inhibitor of AHR (Sasaki-Kudoh 2018), Paclitaxel increases AHR levels (Gao et al 2017) and Adriamycin (doxorubicin) is an activator of AHR (Volkova et al 2011). This adds another perspective to the potential use of these drugs in combination with AHR inhibitors for cancer treatment.
Author Response
Reviewer #2
We would like to thank the reviewers for their careful analysis of our manuscript entitled “How the AHR Became Important in Cancer: The Role of Chronically Active AHR in Cancer Aggression” by Wang et al. We feel that the comments were fair and insightful and significantly helped us improve the manuscript. We have addressed the reviewers’ comments below. Again, thank you for your consideration and input.
The paper by Wang et al “How the AHR Became Important in Cancer: The Role of Chronically Active AHR in Cancer Aggression“ is, in a way, a non-traditional paper containing elements of both review and original research articles. Contrary to a usual review, it contains e.g. a section “Materials and Methods”. This hybrid nature is not a bad thing in itself, but it gives a reviewer challenge to keep these two parts separate.
As noted above, we have seen and written reviews that include primary data so there are precedents for this approach to a review paper. The editors of IJMS have agreed that inclusion of primary data is appropriate when it “…helps the flow of the review or reinforces the summary conclusions in the review”. Our intention was to use these data to help illustrate the main conclusions of the review. For example, Figure 2 assembles publically available data to illustrate the conclusion of many studies that the AHR is active (nuclear) in several cancers. Similarly, several published studies suggest that AHR activity is likely to have an adverse functional consequence in some but perhaps not all cancers. Figure 3, the presentation of survival data for patients with the cancers called out in Figure 2 and expressing higher levels of the AHR transcriptional profile, illustrates this position. That said, we take the reviewers point that it might be confusing to have both a review and primary data. Therefore, we have removed the original Figure 5 and have clarified when we are presenting primary data (e.g., lines 145-148). If the reviewer feels strongly about not having the primary data in this review, then we will gladly remove them.
A review of AHR is a most timely and needed event – over the last couple of years, many interesting original research papers have been published, which give a good reason for a review. In the context of AHR and cancer, there are couple of such reviews published recently (Baker et al Med Res Rev. 2020; Trikha and Lee, Biochim Biophys Acta Rev Cancer. 2020; Kolluri et al Arch Toxicol. 2017) and it would have been nice to see them mentioned by Wang et al.
We agree and apologize for having not included these reviews in the first submission. We now have referenced the Baker et al and the Trikha et al manuscripts in Section 2 where the role of the AHR in immune system development is noted and the Kolluri et al manuscript in Section 6.3, lines 381-388 in the context of paradoxical effects of the AHR in cancer.
AHR is a transcription factor and thus a very important part of cell signaling pathway. It is activated by ligands (exo- and endogenous), after which AHR translocates into nucleus and dimerizes with ARNT. It would be useful to start the review from this known molecular signaling pathway, because it is very central in explaining the functioning of AHR in both normal and cancer cells.
As suggested, we now have added a new Section 2 and Figure (Figure 1) that describes AHR signaling and references to related review articles (for example, reference 39).
Line 51 speaks about “induction of hydroxylases“ and line 55 about “AHR-mediated transcriptional up-regulation” - these expressions are simply not clear enough. And, after all, ARNT comes in as an AHR partner only at line 180, when the hypothetical model of AHRR action is described. So, my suggestion would be to start with Fig.1., which could explain the cellular AHR pathway.
We agree and have added a new figure (Figure 1) depicting the canonical AHR signaling pathway with “hydroxylases” explained in the legend.
A problem in this (and not only this) paper is that there is no clear distinction between results achieved with mouse models (and cells) and human ones. Our own experience shows that there are important and substantial differences between the two organisms and putting the data all together sometimes leads to a confusion.
Thank you for raising this important point. We now have added a new section (Section 7) in which we discuss potential interspecies differences.
Over the last two years, more than 100 papers have been published in the AHR field. Although the Wang et al review covers excellently the historical part of AHR research, which started in 1970-ies, it would be important to include also much more data from recent years. Along the same lines – a 2016 paper (Kwon et al) can hardly be called a recent one in a review, which will be published in 2021 (see line 279).
We agree and have added several relevant papers from 2018-2020 including 12 from 2018, 9 from 2019, and 10 from 2020.
I have also some remarks concerning the original research results included in the paper. First, the Affimetrix data of two cell lines with knock-out AHR claim that there were 644 differences (down-regulations?) in gene expression between AHR+ and AHR- lines. For any standard of scientific publication, one would expect to see the list of these 644 genes at least as a Supplementary information or equivalent.
We agree. We now have included a supplementary information Table 1 with the 644 gene biomarker set listed along with their adjusted values from the two cell lines in which the AHR was deleted. We also are depositing the knockout data in GEO.
Additional small technical problem is the use of cells transfected with Cas9 and no guide RNA as a control equivalent to wt. At Fig. 4a one can see that AHR expression is higher in Cas9 transfected cells that in wt, which itself can cause changes in expression levels of some of these 644 genes.
Quantification of band densities from three experiments with SUM149 cells and three experiments with MDA-MB-231 showed no statistical difference between the wildtype and Cas9 AHR levels in either line. This is now more clearly shown in Figure 2 by inclusion of quantification of band densities from three experiments per line.
As correctly stated (lines 79 and further; 105 and others), higher AHR expression levels have been seen in many cancers. However, this is not always the case and there are many examples of the contrary. I find that this controversy would need some discussion, especially in the context of Fig.2.
We agree that this is important point. Therefore, at the end of section 6.3, we have added a discussion of the paradox that AHR agonists and antagonists can appear to have the same beneficial effect in various cancer contexts.
The exception of Twist1 (line 258) in the list of the genes increasing their expression after AHR hyperactivation is not entirely correct – one can find 5’-GCGTG-3’ in both human and mouse Twist1 DNA sequences.
Thank you for correcting that. We have made the change in section 6.2 to include Twist1.
Line 324 and further: AHR inhibitors increase the sensitivity of breast carcinoma cells to Adriamycin and Paclitaxel and oral cancer cells to Cisplatin. It should be mentioned, however, that Adriamycin, Paclitaxel and Cisplatin also interact with AHR directly. Cisplatin is an inhibitor of AHR (Sasaki-Kudoh 2018), Paclitaxel increases AHR levels (Gao et al 2017) and Adriamycin (doxorubicin) is an activator of AHR (Volkova et al 2011). This adds another perspective to the potential use of these drugs in combination with AHR inhibitors for cancer treatment.
This is an excellent point that reminds us that AHR signaling is complicated. We now have addressed the implication of AHR activation by these chemotherapeutics and have acknowledged those papers in Section 6.3, lines 355-366.
Round 2
Reviewer 1 Report
The authors have addressed all the issues and I has no additional comments. This manuscript could be accepted for publication